# Association of Thymidylate Synthase (*TS*) Gene Polymorphisms with Incidence and Prognosis of Coronary Artery Disease

**DOI:** 10.3390/ijms241612591

**Published:** 2023-08-09

**Authors:** Jung Oh Kim, Chang Soo Ryu, Jeong Yong Lee, Eun Ju Ko, Yong Hyun Ha, Jung Hoon Sung, Tae Sun Hwang, In Jai Kim, Nam Keun Kim

**Affiliations:** 1Department of Biomedical Science, College of Life Science, CHA University, Seongnam 13488, Republic of Korea; kjo@basgenbio.com (J.O.K.); regis2040@nate.com (C.S.R.); smilee3625@naver.com (J.Y.L.); ejko05@naver.com (E.J.K.); hayo119@naver.com (Y.H.H.); 2Genetic Epidemiology Research Institute, Basgenbio Inc., Seoul 04167, Republic of Korea; 3Department of Internal Medicine, CHA Bundang Medical Center, CHA University, Seongnam 13496, Republic of Korea; atropin5@cha.ac.kr; 4Department of Anatomy, School of Medicine, CHA University, Seongnam 13488, Republic of Korea; tshwang@cha.ac.kr

**Keywords:** coronary artery disease, thymidylate synthase, 3′-untranslated region, genetic variants, post-transcriptional regulation

## Abstract

Coronary artery disease (CAD) is a prevalent cardiovascular condition characterized by the accumulation of plaque within coronary arteries. While distinct features of CAD have been reported, the association between genetic factors and CAD in terms of biomarkers was insufficient. This study aimed to investigate the connection between genetic factors and CAD, focusing on the thymidylate synthase (*TS*) gene, a gene involved in DNA synthesis and one-carbon metabolism. TS plays a critical role in maintaining the deoxythymidine monophosphate (dTMP) pool, which is essential for DNA replication and repair. Therefore, our research targeted single nucleotide polymorphisms that could potentially impact *TS* gene expression and lead to dysfunction. Our findings strongly associate the *TS* 1100T>C and 1170A>G genotypes with CAD susceptibility. We observed that *TS* 1100T>C polymorphisms increased disease susceptibility in several groups, while the *TS* 1170A>G polymorphism displayed a decreasing trend for disease risk when interacting with clinical factors. Furthermore, our results demonstrate the potential contribution of the *TS* 1100/1170 haplotypes to disease susceptibility, indicating a synergistic interaction with clinical factors in disease occurrence. Based on these findings, we propose that polymorphisms in the *TS* gene had the possibility of clinically useful biomarkers for the prevention, prognosis, and management of CAD in the Korean population.

## 1. Introduction

Coronary artery disease (CAD) is a vascular disorder characterized by ischemia and stenosis [1]. CAD involves the development of atherosclerotic plaques in epicardial coronary arteries [2], leading to the narrowing of the coronary artery lumen and impaired antegrade myocardial blood flow. Patients who have experienced myocardial infarction (MI), undergone percutaneous coronary intervention (PCI), or received a coronary artery bypass graft are diagnosed with coronary heart disease (CHD) [3]. CAD remains the leading cause of morbidity and mortality worldwide, with an estimated 30% of adults affected by its long-term consequences. Considering the aging population and the increasing prevalence of risk factors such as obesity and diabetes, it is projected that more than 23.3 million people worldwide will succumb to acute MI, stroke, and CAD annually by 2030 (World Health Organization, 2008) [4].

In Korea, CAD exhibits a high incidence and mortality rate, prompting numerous studies on this disease [2,3,4,5]. While previous studies have proposed various causes, treatment approaches, and prognostic management strategies for CAD [5,6,7,8], the exact etiology of the disease remains elusive. Moreover, CAD manifests with varying severity, onset times, treatment responses, and prognostic outcomes among individuals, posing challenges in managing disease prognosis. Consequently, improved diagnostic methods for early CAD detection and identification of at-risk populations are crucially needed.

CAD is a complex, multifactorial, and polygenic disorder resulting from interactions between various genes and environmental factors. Several factors, such as hypertension, diabetes mellitus (DM), smoking, hyperlipidemia, and hyperhomocysteinemia, are associated with an increased risk of CAD [5]. Notably, hyperhomocysteinemia is recognized as an independent and potentially modifiable risk factor for vascular diseases. This association has been reported in numerous studies involving diverse ethnic groups [9,10,11]. Given this background, our study aimed to investigate the relationship between genetic variants of thymidylate synthase (*TS*), a key factor in homocysteine (Hcy) and folate metabolism, and CAD.

TS catalyzes the reductive methylation of deoxyuridine monophosphate (dUMP) by folate to produce deoxythymidine monophosphate (dTMP). It has been extensively studied in terms of its structure, function, and inhibition [12]. Typically, TS exists as a dimer composed of identical 30–35 kDa subunits. The enzyme catalyzes the reductive transfer of the methylene group from 5,10-methylene-tetrahydrofolate (5,10-MTHF) to the 5′-position of the substrate deoxyuridylic acid to form TMP and dihydrofolate (DHF) [12]. TS plays a critical role in the proliferation of cells and serves as a target for various chemotherapeutic drugs that mimic either the substrate or cofactor [13].

The *TS* gene belongs to the S-phase gene family, whose expression is significantly upregulated at the G1/S-phase boundary following the initiation of DNA replication. The expression of this family of genes may be coordinated through a common factor or mechanism [13,14]. Recent studies have suggested that the transcription of several S-phase genes, including *DHFR* and *TK*, may be partially controlled at the transcriptional level by the E2F family of transcription factors [15]. Previous investigations on *TS* mRNA regulation have focused on the insertion/deletion of nucleotides in the promoter region. Additionally, transcriptional control represents only one aspect of the regulatory mechanisms influencing the expression of numerous S-phase genes. Other levels of control include RNA processing, mRNA translation, mRNA stability, and protein stability. Each gene appears to be regulated through a unique combination of mechanisms. Despite these studies, the regulatory mechanisms of the *TS* gene remain largely unclear.

Therefore, our study aimed to explore disease-related *TS* gene polymorphisms. We investigated the association between vascular diseases, including CAD, and the *TSER* 2R/3R and *TS* 3′-UTR variants (*TS* 1100T>C, *TS* 1170A>G, and *TS* 1494ins/del). Furthermore, we analyzed the differences based on genetic variants in the *TS* 3′-UTR, focusing on the synergic effect of the clinical factors of CAD.

## 2. Results

### 2.1. Clinical Profiles of CAD Patients and Control Subjects

The characteristics of CAD patients and control subjects are presented in Table 1. The CAD patient group and control group consisted of 35.8% and 36.1% males, respectively. The mean age for CAD patients and control subjects was 62.55 ± 10.26 years and 61.44 ± 11.52 years, respectively. Significant differences were observed in clinical factors between the CAD patients and control subjects. Specifically, there were significant differences in the levels of metabolic syndrome, BMI, hypertension, diabetes, fasting blood sugar, hyperlipidemia, HDL-C, total cholesterol, and triglyceride when comparing clinical indicators between CAD patients and control patients (*p <* 0.05).

### 2.2. Genotype Frequencies of TS Gene Polymorphisms in CAD Patients and Control Subjects

We focused our investigations on the *TSER* 2R/3R, *TS* 1100T>C, *TS* 1170A>G, and *TS* 1494ins/del polymorphisms. Table 2 displays the genotype distributions in CAD patients and control subjects. AORs were calculated using logistic regression analysis, considering age, sex, hypertension, DM, smoking, and hyperlipidemia. The genotype frequencies of *TS* in the control group were consistent with the Hardy–Weinberg equilibrium expectations.

Significant differences were observed in the *TS* gene 3′-UTR polymorphisms *TS* 1100T>C and *TS* 1170A>G between CAD patients and control subjects. In CAD patients, the *TS* 1100T>C polymorphism (TT versus TC+CC: AOR = 1.350, 95% CI = 1.014–1.797, *p* = 0.040) was significantly associated with an increased risk of disease compared to the control group. Conversely, the *TS* 1170A>G polymorphism showed a decreased occurrence risk in CAD patients (AA versus GG: AOR = 0.464, 95% CI = 0.242–0.889, *p* = 0.021; AA versus AG versus GG: AOR = 0.763, 95% CI = 0.602–0.966, *p* = 0.025; AA+AG versus GG: AOR = 0.532, 95% CI = 0.288–0.983, *p* = 0.044) compared to the control group. Consistent with the results of the CAD, the association of the *TS* 1100T>C and *TS* 1170A>G polymorphisms with CAD is an interesting finding.

Furthermore, we identified the genotype frequencies of *TS* gene 3′UTR polymorphisms between stable CAD (205 individuals, 48.3%), acute coronary syndrome (ACS; 219 individuals, 51.7%), and control groups. Significant differences were observed in the *TS* 1170A>G polymorphism between ACS patients and control subjects (Appendix A).

### 2.3. Haplotype and Genotype Combination Analysis of TSER and TS 3′-UTR Gene Polymorphisms between CAD Patients and Control Subjects

The linkage disequilibrium of *TS* polymorphisms at loci *TSER* (rs45445694), *TS* 1100 (rs699517), *TS* 1170 (rs2790), and *TS* 1494 (rs151264360) was examined in CAD patients and control subjects (Figure 1 and Appendix A). Strong linkage disequilibrium was observed between loci *TS* 1100 and *TS* 1170 (D′ = 0.936, LOD = 111.45, r2 = 0.742) between the CAD and control groups (Figure 1).

Haplotype analysis was performed for four loci (*TSER*/*TS* 1100/*TS* 1170/*TS* 1494), three loci (*TSER*/*TS* 1100/*TS* 1170, *TSER*/*TS* 1100/*TS* 1494, and *TSER*/*TS* 1170/*TS* 1494), and two loci (*TSER*/*TS* 1100, *TSER*/*TS* 1170, *TSER*/*TS* 1494, *TS* 1100/*TS* 1170, *TS* 1100/*TS* 1494, and *TS* 1170/*TS* 1494) associated with CAD, as presented in Table 3 and Appendix A, respectively. Among the four locus *TS* haplotypes, the 3R-T-G-0bp and 3R-C-A-0bp haplotypes were commonly found in CAD patients. The 3R-T-G-0bp haplotype significantly reduced the risk of CAD (OR = 0.699, 95% CI = 0.546–0.896, *p* = 0.006), while the 3R-C-A-0bp haplotype significantly increased the risk of disease (OR = 19.690, 95% CI = 4.727–81.990, *p <* 0.0001). In the analysis of three *TS* loci (Table 3), the 3R-C-0bp (*TSER*/*TS* 1100/*TS* 1494) and C-A-0bp (*TS* 1100/*TS* 1170/*TS* 1494) haplotypes were associated with CAD occurrence (OR = 16.522, 95% CI = 5.110–53.414, *p <* 0.0001; OR = 20.462, 95% CI = 4.918–85.133, *p <* 0.0001), while the 3R-T-G, 2R-T-A (*TSER*/*TS* 1100/*TS* 1170), and T-G-0bp (*TS* 1100/*TS* 1170/*TS* 1494) haplotypes were significantly associated with a reduced risk of CAD. Interestingly, in the haplotype analysis of two *TS* loci (Table 3), the 3R-G (*TSER*/*TS* 1170), T-G (*TS* 1100/*TS* 1170), C-0bp (*TS* 1100/*TS* 1494), and G-0bp (*TS* 1170/*TS* 1494) haplotypes showed a significant association with CAD risk. The C-0bp haplotype (*TS* 1100/*TS* 1494) had the highest risk effect on CAD occurrence (OR = 16.551, 95% CI = 5.122–53.488, *p <* 0.0001), while the G-0bp haplotype (*TS* 1170/*TS* 1494) was most protective against CAD incidence (OR = 0.671, 95% CI = 0.530–0.851, *p* = 0.001). These results of haplotype analysis also showed a continuing significant association with CAD risk in the FDR *p*-value.

Genotype combination analysis was performed for *TSER* 2R/3R, *TS* 1100T>C, *TS* 1170A>G, and *TS* 1494ins/del polymorphisms, and these results are shown in Appendix A. Among genotype combination analyses, many kinds of genotype combinations in *TSER*/*TS*1170, *TS* 1100/*TS* 1170, and *TS* 1170/*TS* 1494 led to a decrease in CAD risk (Table 4). In contrast, the TC/0bp0bp genotype combination in *TS* 1100/*TS* 1494 (Table 4) is associated with a highly increased CAD risk (AOR = 26.713, 95% CI = 3.462–206.115, *p* = 0.002). Interestingly, this *TS* 1100/1494 haplotype result is maintained in the *TS* 1100/*TS* 1494 genotype combination.

### 2.4. Combined Effects between TS Gene Polymorphisms and Environmental Factors on CAD Prevalence

We conducted stratified analyses to investigate the association between *TS* polymorphisms and CAD, taking into account various clinical factors, including age, sex, hypertension, DM, hyperlipidemia, smoking status, plasma Hcy, and folate levels (Appendix A). The *TS* 1100T>C polymorphism was associated with increased disease risk in the smoking group (CC versus CA+AA: AOR = 2.016, 95% CI = 1.159–3.507, *p* = 0.013). On the other hand, the *TS* 1170A>G polymorphism was associated with a lower risk of disease in the low Hcy group (AA+AG versus GG: AOR = 0.157, 95% CI = 0.052–0.471, *p* = 0.001).

We examined the interactions between the *TS* gene polymorphisms and environmental factors in the context of CAD (Figure 2 and Appendix A). The combined effect of clinical factors and genetic polymorphisms significantly contributed to CAD. A synergistic effect was observed with plasma folate levels. In individuals with folate levels < 4.85 ng/mL, there was a synergistic effect between *TSER* 3R2R+2R2R and *TS* 1100TC+CC (AOR = 2.250, 95% CI = 1.243–4.071; AOR = 2.621, 95% CI = 1.585–4.333), while *TS* 1170 showed a synergistic effect in individuals with folate levels ≥ 4.85 ng/mL (AOR = 0.697, 95% CI = 0.501–0.970). A synergistic effect with BMI values was observed in the CAD group (Figure 2). The presence of *TSER* 2R/3R or *TS* 1100T>C polymorphisms in individuals with high BMI values was associated with elevated CAD risk (*TSER* 2R2R+2R3R: AOR = 4.210; *TS* 1100TC+CC: AOR = 4.340) (Figure 2A,B). Furthermore, *TS* 1100TC+CC in individuals with DM was associated with elevated CAD risk (AOR = 3.996) (Figure 2C). Additionally, other *TS* gene polymorphisms (*TSER* 3R2R+2R2R, *TS* 1100TC + CC, *TS* 1170AG + GG, and *TS* 1494 0bp6bp + 6bp6bp) showed synergistic effects with multiple clinical factors (Appendix A).

### 2.5. Baseline Characteristics and Genotype Frequencies in CAD Patients and Control Subjects Stratified by Replication Groups

Baseline characteristics and genotype frequencies were compared between CAD patients and control subjects in two replication groups. A total of 844 individuals (CAD patients and control subjects) from case–control sample 1 (recruitment period: 2000 to 2006) and sample 2 (recruitment period: 2007 to 2012) were included in the analysis (Appendix A). Appendix A presents the baseline characteristics of CAD patients and control subjects in both sample groups. The CAD patients showed higher frequencies of MetS (*p <* 0.0001, sample 1; *p <* 0.0001, sample 2), DM (*p <* 0.0001, sample 1; *p* = 0.020, sample 2), and hypertension (*p* = 0.048, sample 1; *p* = 0.178, sample 2) compared to the controls (Appendix A). Additionally, the genotype frequencies of the *TS* polymorphisms were significantly different between the control and CAD groups in sample 1 (Appendix A).

## 3. Discussion

The mortality rate associated with CAD is nearly three times higher than that of a stroke, underscoring the importance of screening for CAD and identifying diagnostic markers to improve prognosis [16,17,18]. Moreover, numerous studies have investigated the prevalence of subclinical cardiovascular disease in ischemic stroke patients using coronary computed tomographic angiography (CTA) and various surrogate markers of systemic atherosclerosis. These studies have explored the relationship between subclinical CAD and vascular risk factors [19]. Therefore, we aimed to identify associations between CAD onset and diagnostic markers. To address this objective, we recognized the necessity to investigate the onset and treatment of the disease. Based on this rationale, we comprehensively analyzed mutations in the *TS* gene in CAD patients and control subjects.

A recent study reported that the *TSER* 3R allele is associated with increased *TS* expression levels [20]. Therefore, it is important to understand how elevated *TS* expression, as influenced by 3′-UTR polymorphisms, can contribute to the occurrence and prognosis of CAD. *TS* plays a critical role in Hcy and folate metabolism, and genetic variations in enzymes involved in this pathway can influence an individual’s susceptibility to disease [21]. Studies have suggested that elevated *TS* expression can lead to increased Hcy levels and decreased folate levels, which in turn contribute to ischemia development [21,22]. Furthermore, plasma folate concentrations are inversely correlated with Hcy levels [23]. The role of hyperhomocysteinemia in vascular and thromboembolic disease has been extensively studied and debated, with significant vascular disease observed in individuals with markedly elevated plasma Hcy [24,25,26]. Elevated plasma Hcy is thought to increase the risk of thrombosis by causing endothelial injury in both venous and arterial vasculature [25]. Additionally, folate is essential for the de novo synthesis of purines and thymidylate, which are required for DNA replication and repair [27]. Abnormal folate status is implicated in various diseases, including cardiovascular disease, neural tube defects, cleft lip and palate, late pregnancy complications, as well as neurodegenerative and psychiatric disorders [28].

Polymorphisms in the 3′-UTR region of the *TS* gene can potentially affect mRNA stability and translation, leading to significant changes in gene expression. These polymorphisms can either abolish, weaken, or create binding sites for miRNAs, thereby modulating their binding activity. However, there is currently limited data available regarding the modulation of miRNA binding activity based on *TS* 3′-UTR polymorphisms. A study investigating the association between *TS* 1170A>G polymorphism and coronary heart disease risk identified miR-215 and miR-192 as potential miRNAs with binding activity [29]. This suggests that miRNAs may play a crucial role in the prevalence and progression of cardiovascular diseases, as their expression can be altered in specific genotypes [30,31,32,33]. To further understand the impact of these polymorphisms, it will be necessary to investigate the binding activity of miRNAs directly on *TS* 3′-UTR polymorphisms. This investigation will shed light on how these polymorphisms may influence cellular proliferation and the progression of ischemic events. Such studies may hold significant clinical implications for diseases associated with one-carbon metabolism.

In our study, we aimed to investigate the association between four *TS* gene polymorphisms located in the enhancer region and miRNA binding site (3′-UTR) with the prevalence and prognosis of CAD. There was a strong association between the *TS* 1100T>C and *TS* 1170A>G genotypes and susceptibility to CAD. These polymorphisms were also effective predictors of poor prognosis. Moreover, there was a synergistic effect between the *TS* 1170A>G polymorphism and other risk factors on CAD incidence. We observed elevated CAD prevalence when considering interactions between *TSER* and *TS* 1100T>C polymorphisms with environmental factors. Furthermore, specific haplotypes involving the *TS* 1100C and *TS* 1494 insertion alleles were significantly associated with increased CAD incidence, while the combination of the *TS* 1170G allele and *TS* 1494 insertion allele decreased CAD occurrence. To our knowledge, this is the first study providing evidence of an association between 3′-UTR polymorphisms of *TS* and susceptibility to CAD and its progression. Interestingly, despite the *TS* 1100C and *TS* 1170G alleles being located only 70 bp apart within the same gene, our association study revealed conflicting results in terms of their genotype effects. The *TS* 1100CC genotype was significantly associated with increased CAD incidence in our analysis, while the *TS* 1170GG genotype was associated with decreased CAD occurrence.

TS enzyme levels exhibit a significant increase in rapidly proliferating cells compared to resting cells. When resting cells are stimulated to proliferate, TS activity remains unchanged until DNA replication begins, at which point it increases by at least 10-fold during the S-phase [34]. *TS* mRNA content also shows a 10-fold increase as cells progress from G0 through S-phase. However, nuclear run-on transcription assays indicate minimal changes in *TS* gene transcription during the G1-S transition [35,36]. This suggests that regulation of *TS* mRNA primarily occurs at the post-transcriptional level in human and mouse cells undergoing growth stimulation. The half-life of poly(A)+ *TS* mRNA is approximately 8 h in both resting and growing mouse cells [37,38,39], suggesting that mRNA stability regulation is not a critical factor. Thus, *TS* mRNA dysregulation profoundly affects cell proliferation and apoptosis, potentially leading to abnormalities in vascular endothelial cells and compromised blood vessel function.

Endothelial dysfunction initiates a detrimental cycle culminating in overt atherosclerosis, significant CAD, silent brain infarction (SBI), plaque rupture, and ultimately MI or ischemic stroke [40,41,42]. In addition to classic risk factors like hypertension, smoking, DM, and hypercholesterolemia, physical inactivity has emerged as an independent predictor for CAD development [22,43,44,45,46]. Therefore, the identification of genetic diagnostic markers for CAD and a comprehensive understanding of its underlying causes is crucial for effective disease management. However, the study of *TS* genes in the context of vascular disease remains largely unexplored. The *TS* gene has been extensively studied, with a focus on its implications in cancer incidence and treatment [47,48,49,50,51,52,53,54]. Particularly, there is ongoing research on pharmacogenetic activities for the development of anticancer drugs. However, limited attention has been given to the role of the *TS* gene in vascular-related diseases such as ischemia, thrombosis, and stenosis. Therefore, this study represents the first report on the association between the *TS* gene and the pathogenesis of specific vascular diseases, specifically CAD.

There are several limitations to this study. Firstly, the exact mechanism by which 3′-UTR polymorphisms in the *TS* gene influence CAD development remains unclear and warrants further investigation. Secondly, the control group in our study compromised individuals who sought medical attention, which may introduce biases. Future studies should consider recruiting a healthier control group with comprehensive imaging and laboratory tests to minimize potential biases in vascular factor assessment. Thirdly, our study focused on the Korean population, and therefore, the generalizability of our findings to other ethnic groups may be limited. To validate the potential of 3′-UTR variants in the *TS* gene as biomarkers for CAD prevention and prognosis, a larger prospective study involving diverse populations is necessary.

## 4. Materials and Methods

### 4.1. Study Approval and Population

The study protocols were reviewed and approved by the Institutional Review Board of CHA Bundang Medical Center in June 2000, adhering to the principles of the Declaration of Helsinki. This study recruited participants from the South Korean provinces of Seoul and Gyeonggi-do between 2000 and 2012. Informed consent was obtained from all study participants.

A total of 424 consecutive CAD patients referred from the Department of Cardiology at CHA Bundang Medical Center, CHA University, were included in this study. These patients presented with stable CAD or acute coronary syndromes, including unstable angina with or without ST-segment elevation, and had at least one coronary lesion with more than 50% stenosis in a vessel with a diameter of 2.25–4.00 mm. The screening for eligibility occurred between 2006 and 2012. There were no restrictions on the number of treated lesions, treated vessels, lesion length, or the number of stents implanted. Exclusion criteria included acute MI and a life expectancy of less than 1 year. All patients underwent coronary angiography and electrocardiography for diagnosis, which was confirmed by at least one independent, experienced cardiologist.

Additionally, we selected 427 control subjects who were sex- and age-matched (±5 years) to the patient group. These control subjects were patients presenting at our hospitals during the same period for health examinations, including biochemical testing, electrocardiogram, coronary computed tomography (CT), and brain magnetic resonance imaging (MRI). Control subjects had no recent history of anginal symptoms, cerebrovascular disease, or MI. The same exclusion criteria used for the patient group were applied to the control subjects. Hypertension was defined as systolic pressure > 140 mm Hg and diastolic pressure > 90 mm Hg on more than one occasion, including patients currently taking hypertensive medications. DM was defined as a fasting plasma glucose level > 126 mg/dL (7.0 mmol/L) and included patients taking diabetic medications. Smoking was defined as patients who were current smokers. Hyperlipidemia was defined as a high fasting serum total cholesterol level (≥240 mg/dL) or a history of treatment with an antihyperlipidemic agent. The number of control subjects (427) differs from the patient group (424) due to availability and matching criteria.

### 4.2. Assessment of Biochemical Factors

Plasma samples were collected within 49 h of stroke onset to measure the levels of total Hcy and folate. Whole blood was collected from patients 12 h after their last meal using tubes containing anticoagulants. Tubes were centrifuged for 15 min at 1000× *g* to separate the plasma. Total plasma Hcy concentrations were measured using a fluorescent polarizing immunoassay with an IMx system (Abbott Laboratories, Chicago, IL, USA), while plasma folate concentrations were measured using an immunoassay kit (ACS 180; Bayer, Tarrytown, NY, USA). Levels of high-density lipoprotein-cholesterol (HDL-C) were determined using enzymatic colorimetric methods with commercial reagent sets (TBA 200FR NEO, Toshiba Medical Systems, Tokyo, Japan).

### 4.3. Genotyping

DNA extraction from leukocytes was conducted using a G-DEX II Genomic DNA Extraction kit (Intron Biotechnology, Seongnam, Republic of Korea) according to the manufacturer’s instructions. Genotyping of the TS gene was performed using the polymerase chain reaction–restriction fragment length polymorphism (PCR–RFLP) method, which is a cost-effective alternative to whole-genome sequencing. The *TSER* 2R/3R (28 bp tandem repeat) was detected using a forward primer (5′-GTG GCT CCT GCG TTT CCC CC-3′) and a reverse primer (5′-GCT CCG AGC CGG CCA CAG GCA TGG CGC GG-3′). The resulting PCR products of 248 bp and 220 bp were then digested with 5U Hae III. The presence of a 113 bp fragment indicated the 2R2R genotype, while fragments of 66 bp, 47 bp, and 28 bp indicated the 2R3R genotype, and fragments of 94 bp, 47 bp, and 28 bp indicated the 3R3R genotype. The *TS* gene polymorphisms (*TSER* 2R/3R, *TS* 1100T>C, and *TS* 1494 ins/del) did not show different frequencies by the ethnic population, but *TS* 1170A>G polymorphism showed the difference by the ethnic population (https://www.ncbi.nlm.nih.gov/snp/, accessed on 4 August 2023).

For the *TS* 1100T>C and *TS* 1170A>G genotypes, PCR–RFLP analysis was conducted using the forward primer (5′-GGT ACA ATC CGC ATC CAA CTA TTA-3′) and reverse primer (5′-CTG ATA GGT CAC GGA CAG ATT T-3′). The amplified fragment had a length of 170 bp and was digested with 5U Ban II (TS 1100T>C) or 3U Mbo II (TS 1170A>G) for 16 h at 37 °C. The presence of a 170 bp fragment indicated the TT genotype for 1100T>C, while fragments of 170 bp, 108 bp, and 62 bp indicated the TC genotype, and fragments of 108 bp and 62 bp indicated the CC genotype. For *TS* 1170A>G, the presence of a 170 bp fragment indicated the AA genotype, fragments of 170 bp, 142 bp, and 28 bp indicated the AG genotype, and fragments of 142 bp and 28 bp indicated the GG genotype.

The *TS* 1494ins/del polymorphism was detected using PCR–RFLP analysis with the forward primer (5′-CAA ATC TGA GGG AGC TGA GT-3′) and reverse primer (5′-CAG ATA AGT GGC AGT ACA GA-3′). The resulting 158 bp product was digested with 5U Dra I for 16 h at 37 °C. The presence of a 158 bp fragment indicated the 0bp0bp genotype, fragments of 158 bp, 88 bp, and 70 bp indicated the 0bp6bp genotype, and fragments of 88 bp and 70 bp indicated the 6bp6bp genotype.

To validate the PCR–PFLP findings, 30% of the PCR assays were randomly selected and repeated. The repeated samples underwent DNA sequencing using an ABI 3730xl DNA Analyzer (Applied Biosystems, Foster City, CA, USA). The quality control samples demonstrated a 100% concordance rate.

### 4.4. Statistical Analysis

Baseline characteristics were analyzed using chi-square tests for categorical data and Student’s *t*-tests for continuous data to compare patient and control baseline data. The associations between *TS* polymorphisms and CAD incidence were estimated using adjusted odds ratios (AORs) and 95% confidence intervals (95% CIs) through multivariate logistic regression. Regression models were adjusted for age, gender, hypertension, DM, hyperlipidemia, and smoking status, as these classical risk factors for vascular abnormalities are commonly associated with CAD. GraphPad Prism 4.0 (GraphPad Software Inc., San Diego, CA, USA) and Medcalc version 12.7.1.0 (Medcalc Software, Mariakerke, Belgium) were used for statistical analyses.

Haplotypes for multiple loci were estimated using the expectation–maximization algorithm with SNPAlyze (Version 5.1; DYNACOM Co, Ltd., Yokohama, Japan). The association between *TS* gene polymorphisms and long-term prognosis after ischemic stroke was evaluated by tracking survival time from stroke onset to death. The *p*-values of the false discovery rate (FDR) were calculated when performing multiple comparisons to estimate the overall experimental error rate resulting from false-positive results [55,56]. Consequently, *p*-values < 0.05 were considered statistically significant.

## 5. Conclusions

In conclusion, our study investigated the association between *TSER* 3R/2R, *TS* 1100T>C, *TS* 1170A>G, and *TS* 1494ins/del polymorphisms and the incidence and progression of CAD. Our findings demonstrated a positive correlation between these genetic variants and the occurrence of unfavorable CAD prognosis, particularly in the presence of common vascular disease risk factors such as hypertension, DM, HDL-C, Hcy, and folate. These results suggest that these *TS* gene polymorphisms may serve as potential biomarkers for predicting CAD susceptibility and prognosis. Further studies are needed to elucidate the underlying mechanisms and validate the clinical utility of these genetic markers in CAD prevention and personalized treatment strategies.

## Figures and Tables

**Figure 1 ijms-24-12591-f001:**
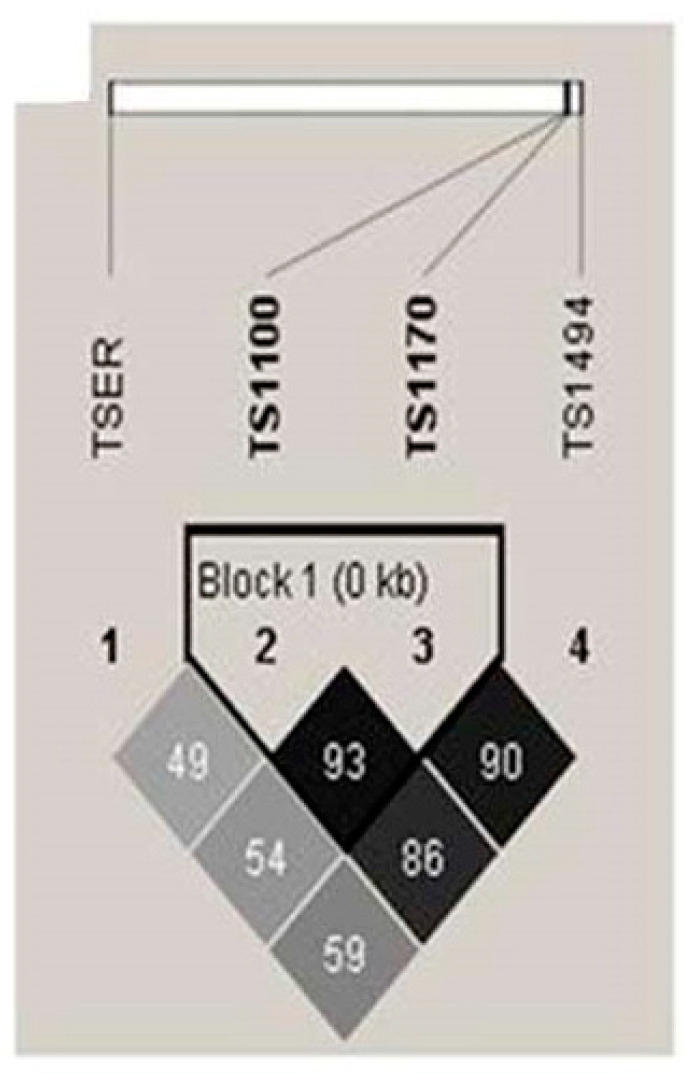
LD patterns of *TS* gene polymorphisms. The values in the squares denote LD between single markers. The association analysis for LD block in control subjects and CAD patients exhibited strong LD block for *TS* 1100/1170. Dark squares indicate high r2 values, and light squares indicate low r2 values. LD, linkage disequilibrium; *TS*, thymidylate synthase; CAD, coronary artery disease.

**Figure 2 ijms-24-12591-f002:**
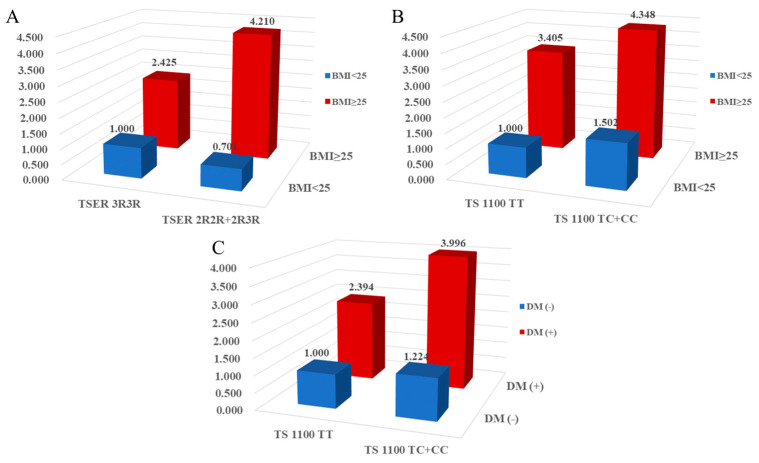
The synergic effect analysis for interplay between clinical factors (body mass index values, diabetes mellitus) and *TS* polymorphisms in coronary artery disease prevalence. (**A**) *TSER* 2R/3R with body mass index (BMI), (**B**) *TS* 1100T>C polymorphism with BMI value, and (**C**) *TS* 1100T>C polymorphism with diabetes mellitus (DM).

**Table 1 ijms-24-12591-t001:** Baseline characteristics between controls and CAD patients.

Characteristic	Controls	CAD Patients	*p*
	(n = 427)	(n = 424)	
Male (n, %)	153 (35.8)	171 (40.3)	0.177
Age (years, mean ± SD)	61.44 ± 11.52	62.55 ± 10.26	0.136
Hypertension (n, %)	171 (40.0)	226 (57.4)	<0.0001
Diabetes mellitus (n, %)	51 (11.9)	119 (30.2)	<0.0001
Hyperlipidemia (n, %)	97 (22.7)	116 (29.4)	0.028
Smoking (n, %)	136 (31.9)	101 (25.6)	0.049
Body mass index (kg/cm^2^, mean ± SD)	24.26 ± 3.34	25.21 ± 3.12	<0.001
Fasting blood sugar (mg/dL, mean ± SD)	111.98 ± 33.93	141.23 ± 63.78	<0.001
HbA1c (%, mean ± SD)	6.08 ± 1.16	6.67 ± 2.99	0.048
HDL-C (mg/dL, mean ± SD)	47.02 ± 13.90	43.83 ± 11.58	0.005
LDL-C (mg/dL, mean ± SD)	121.31 ± 43.32	111.87 ± 38.20	0.012
Total cholesterol (mg/dL, mean ± SD)	194.02 ± 38.14	185.98 ± 45.06	0.006
Triglyceride (mg/dL, mean ± SD)	142.16 ± 86.73	155.52 ± 97.04	0.037
Hcy (μmol/L, mean ± SD)	9.77 ± 3.91	9.92 ± 5.09	0.620
Folate (nmol/L, mean ± SD)	9.14 ± 7.94	8.69 ± 9.73	0.467

*p-*values were calculated by a two-sided *t*-test for continuous variables and a chi-square test for categorical variables. HDL-C, high-density lipoprotein-cholesterol; LDL-C, low-density lipoprotein-cholesterol; Hcy, homocysteine; HbA1c, hemoglobin A1c.

**Table 2 ijms-24-12591-t002:** Genotype frequencies of *TS* gene polymorphisms between CAD patients and control subjects.

Genotypes	Controls	CAD	AOR (95% CI) *	*p* ^a^	FDR-*p*
	(n = 427)	(n = 424)			
*TSER*					
3R3R	296 (69.3)	301 (71.0)	1.000 (reference)		
2R3R	123 (28.8)	115 (27.1)	0.934 (0.677–1.288)	0.676	0.676
2R2R	8 (1.9)	8 (1.9)	1.139 (0.401–3.236)	0.807	0.807
Additive model			0.964 (0.725–1.281)	0.800	0.800
Dominant model			0.945 (0.691–1.293)	0.724	0.724
Recessive model			1.141 (0.398–3.271)	0.806	0.826
HWE-*p*	0.240	0.431			
*TS* 1100T>C					
TT	217 (50.8)	194 (45.8)	1.000 (reference)		
TC	177 (41.5)	189 (44.6)	1.333 (0.989–1.798)	0.060	0.240
CC	33 (7.7)	41 (9.7)	1.344 (0.786–2.298)	0.281	0.562
Additive model			1.249 (1.000–1.562)	0.050	0.100
Dominant model			1.350 (1.014–1.797)	0.040	0.158
Recessive model			1.230 (0.738–2.052)	0.427	0.826
HWE-*p*	0.709	0.607			
*TS* 1170A>G					
AA	211 (49.4)	234 (55.2)	1.000 (reference)		
AG	183 (42.9)	172 (40.6)	0.832 (0.618–1.121)	0.227	0.454
GG	33 (7.7)	18 (4.2)	0.464 (0.242–0.889)	0.021	0.084
Additive model			0.763 (0.602–0.966)	0.025	0.100
Dominant model			0.773 (0.580–1.030)	0.079	0.158
Recessive model			0.532 (0.288–0.983)	0.044	0.176
HWE-*p*	0.439	0.084			
*TS* 1494ins/del					
0bp0bp	195 (45.7)	211 (49.8)	1.000 (reference)		
0bp6bp	193 (45.2)	174 (41.0)	0.907 (0.673–1.223)	0.522	0.676
6bp6bp	39 (9.1)	39 (9.2)	0.878 (0.521–1.480)	0.625	0.807
Additive model			0.934 (0.749–1.164)	0.542	0.723
Dominant model			0.908 (0.683–1.208)	0.508	0.677
Recessive model			0.946 (0.574–1.557)	0.826	0.826
HWE-*p*	0.372	0.717			

Abbreviations: AOR, adjusted odds ratio; 95% CI, 95% confidence interval; CAD, coronary artery disease; TSER, thymidylate synthase enhancer region; TS, thymidylate synthase; HWE, Hardy–Weinberg equilibrium; FDR, false discovery rate. * The AOR on the basis of risk factors such as age, gender, hypertension, diabetes mellitus, hyperlipidemia, and smoking. ^a^
*p*-value calculated by multivariable logistic regression.

**Table 3 ijms-24-12591-t003:** The haplotype analyses for *TS* gene polymorphisms in CAD patients and control subjects.

Haplotype	Control (2n = 854)	CAD (2n = 848)	OR (95% CI)	*p* ^a^	FDR-*p*
*TSER*/*TS*1100/1170/1494					
3R-T-A-0bp	0.3554	0.3916	1.000 (reference)		
3R-T-G-0bp	0.2780	0.2133	0.699 (0.546–0.896)	0.006	0.039
3R-C-A-0bp	0.0024	0.0512	19.690 (4.727–81.990)	<0.0001	0.001
*TSER*/*TS*1100/1170					
3R-T-A	0.3781	0.4148	1.000 (Reference)		
3R-T-G	0.2794	0.2132	0.695 (0.544–0.888)	0.003	0.021
2R-T-A	0.048	0.0307	0.582 (0.348–0.973)	0.033	0.116
*TSER*/*TS*1100/1494					
3R-T-0bp	0.6336	0.6049	1.000 (Reference)		
3R-C-0bp	0.0037	0.0550	16.522 (5.110–53.414)	<0.0001	0.001
*TSER*/*TS*1170/1494					
3R-A-0bp	0.3592	0.4487	1.000 (Reference)		
3R-A-6bp	0.1965	0.1792	0.731 (0.560–0.954)	0.021	0.053
3R-G-0bp	0.2782	0.2109	0.608 (0.475–0.777)	<0.0001	0.001
2R-A-0bp	0.0355	0.0231	0.539 (0.300–0.967)	0.033	0.053
2R-A-6bp	0.1173	0.1037	0.711 (0.514–0.983)	0.038	0.053
2R-G-6bp	0.0002	0.0075	10.51 (0.589–187.400)	0.037	0.053
*TS*1100/1170/1494					
T-A-0bp	0.392	0.4153	1.000 (Reference)		
T-G-0bp	0.287	0.2326	0.765 (0.602–0.973)	0.028	0.098
C-A-0bp	0.0024	0.0509	20.462 (4.918–85.133)	<0.0001	0.001
*TSER*/*TS* 1170					
3R-A	0.5578	0.6302	1.000 (Reference)		
3R-G	0.2794	0.2153	0.683 (0.543–0.858)	0.001	0.003
2R-A	0.1506	0.1245	0.732 (0.551–0.974)	0.031	0.031
2R-G	0.0122	0.0300	2.228 (1.059–4.688)	0.017	0.026
*TS* 1100/1170					
T-A	0.4259	0.4452	1.000 (Reference)		
T-G	0.2896	0.2353	0.780 (0.616–0.987)	0.038	0.057
C-G	0.0020	0.0100	3.852 (0.813–18.261)	0.023	0.057
*TS* 1100/1494					
T-0bp	0.6790	0.6478	1.000 (Reference)		
C-0bp	0.0037	0.0550	16.551 (5.122–53.488)	<0.0001	0.0003
*TS* 1170/1494					
A-0bp	0.3949	0.4715	1.000 (Reference)		
A-6bp	0.3135	0.2832	0.754 (0.602–0.946)	0.015	0.015
G-0bp	0.2878	0.2313	0.671 (0.530–0.851)	0.001	0.003
G-6bp	0.0038	0.0140	3.370 (0.943–12.042)	0.014	0.015

Abbreviations: OR, odds ratio; 95% CI, 95% confidence interval; CAD, coronary artery disease; TSER, thymidylate synthase enhancer region; TS, thymidylate synthase; FDR, false discovery rate. ^a^
*p*-value calculated by the chi-square test and Fisher’s exact test. The *p*-value ≥ 0.05 was excluded in Table 3.

**Table 4 ijms-24-12591-t004:** Genotype combination analyses for the *TS* gene polymorphisms in CAD patients and control subjects.

Genotype Combinations	Controls	CAD	AOR (95% CI)	*p* *
	(n = 427)	(n = 424)		
*TSER*/*TS* 1170A>G				
3R3R/AA	130 (30.4)	164 (38.7)	1.000 (reference)	
3R3R/AG	136 (31.9)	122 (28.8)	0.656 (0.461–0.934)	0.020
3R3R/GG	30 (7.0)	15 (3.5)	0.335 (0.162–0.691)	0.003
2R3R/AA	74 (17.3)	65 (15.3)	0.634 (0.405–0.990)	0.045
*TS* 1100T>C/*TS* 1170A>G				
TT/AA	70 (16.4)	86 (20.3)	1.000 (reference)	
TT/AG	114 (26.7)	92 (21.7)	0.593 (0.376–0.936)	0.025
TT/GG	33 (7.7)	16 (3.8)	0.330 (0.153–0.708)	0.004
*TS* 1100T>C/*TS* 1494ins>del				
TT/0bp0bp	194 (45.4)	180 (42.5)	1.000 (reference)	
TC/0bp0bp	1 (0.2)	21 (5.0)	26.713 (3.462–206.115)	0.002
*TS* 1170A>G/*TS* 1494ins>del				
AA/0bp0bp	57 (13.3)	101 (23.8)	1.000 (reference)	
AA/0bp6bp	117 (27.4)	97 (22.9)	0.479 (0.303–0.755)	0.002
AA/6bp6bp	37 (8.7)	36 (8.5)	0.454 (0.241–0.857)	0.015
AG/0bp0bp	105 (24.6)	94 (22.2)	0.463 (0.292–0.734)	0.001
AG/0bp6bp	76 (17.8)	75 (17.7)	0.539 (0.331–0.876)	0.013
GG/0bp0bp	33 (7.7)	16 (3.8)	0.213 (0.098–0.462)	0.0001

Abbreviation: AOR, adjusted odds ratio; 95% CI, 95% confidence interval; CAD, coronary artery disease; TS, thymidylate synthase. * Adjusted by age, sex, hypertension, diabetes mellitus, hyperlipidemia, and smoking. The *p*-value ≥ 0.05 was excluded in Table 4.

## Data Availability

Not applicable.

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
