# Peer review of "Association of Thymidylate Synthase (TS) Gene Polymorphisms with Incidence and Prognosis of Coronary Artery Disease"

_ijms, 2023, doi:10.3390/ijms241612591_

Round 1

Reviewer 1 Report

The authors investigated the association between TS gene polymorphisms and CAD in 424 patients with CAD and 427 controls. However, I have some comments.

1) How many percentages of patients with CAD were those with stable CAD and those with acute coronary syndrome (ACS), respectively? Moreover, were the genetype frequency of TS gene polymorphisms and its association different between patients with stable CAD and those with ACS?

2) Was the frequency of TS gene polymorphisms in control subjects in this study similar to that reported in Korean or Caucasians general populations?

3) The authors described that this study recruited 424 study patients between 2000 and 2012, that is, only 424 patients during 12 years. Did the authors consecutively recruit patients undergoing coronary angiography or those who visit their outpatient clinic?

4) Was written informed consent obtained from all study subjects?

Author Response

Response to the Reviewer’s comments

Reviewer 1

1) How many percentages of patients with CAD were those with stable CAD and those with acute coronary syndrome (ACS), respectively? Moreover, were the genotype frequency of TS gene polymorphisms and its association different between patients with stable CAD and those with ACS?

=> We thank the reviewer for the critical comments. As you mentioned, we checked the CAD patients, and the stable CAD patients and ACS patients confirmed 48.3%, and 51.7%, respectively. Furthermore, we confirmed the genotype frequency of TS gene polymorphism and its association between patients with stable CAD, ACS, and control subjects. The added sentences and supplementary table are as follows: "Furthermore, we identified the genotype frequencies of TS gene 3’UTR polymorphisms between stable CAD (205 individuals, 48.3%), acute coronary syndrome (ACS; 219 individuals, 51.7%), and control groups. Significant differences were observed in the TS 1170A>G polymorphism between ACS patients and control subjects (Table S9)."

2) Was the frequency of TS gene polymorphisms in control subjects in this study similar to that reported in Korean or Caucasians general populations?

=> We thank the reviewer for the critical comments. As you mentioned, we checked that our frequencies of several TS gene polymorphisms in control subjects were similar to that reported in the Korean or Caucasian general population. The added sentence are as follows: “The TS gene polymorphisms (TSER 2R/3R, TS 1100T>C, and TS 1494 ins/del) weren't showed that different frequencies by the ethnic population, but TS 1170A>G polymorphism showed the difference by the ethnic population (https://www.ncbi.nlm.nih.gov/snp/).”

3) The authors described that this study recruited 424 study patients between 2000 and 2012, that is, only 424 patients during 12 years. Did the authors consecutively recruit patients undergoing coronary angiography or those who visit their outpatient clinic?

=> We thank the reviewer for the valuable comments. As you mentioned, we tried to recruit patients, but it was not enough to meet more specific clinical criteria for research or difficulties in persuading patients to consent to the study. In addition, when considering the QC criteria of DNA for the experiment or the missing variables necessary for the analysis, the analysis sample was finally set to 424 people. The mentioned sentences are as follows: "All patients underwent coronary angiography and electrocardiography for diagnosis, which was confirmed by at least one independent experienced cardiologist."

4) Was written informed consent obtained from all study subjects?

=> We thank the reviewer for the valuable comments. As you mentioned, we already have written informed consent that obtained from all study subjects. Also, we were mentioned the comment of informed consent statement section. The mentioned sentences are as follows: “Written informed consent was obtained from all CAD patients and control subjects.”

Reviewer 2 Report

I read with interest the article by Jung Oh Kim et al.

The authors investigated the possible association of TS polymorphisms with incidence and prognosis of CAD.

I have a few minor and a major remarks.

- In the Abstract, there is no clue on the thought-process behind selecting this particular gene to investigate the association with CAD. Is should be reported why the authors opted to investigate this particular gene among others.

- It is unclear why in the first paragraph, in the Discussion section the authors undertook a discussion around association of CAD and stroke; this is quite irrelevant with the study.

- Endothelial dysfunction is not necessarily a precursor of CAD ('Endothelial dysfunction, a precursor to cardiovascular or cerebrovascular diseases', Discussion section). Endothelial dysfunction could co-exist, predate, antedate or any combination of these.

- Some sections need to be toned down.

Eg the causes of coronary artery disease are far from unclear and hence the sentence 'While distinct features of CAD have been identified, its causes remain largely unclear' should be changed or deleted altogether.

Additionally, bold statements like 'we propose that polymorphisms in 30 the 3'-UTR miRNA binding site of the TS gene could serve as clinically useful biomarkers for the 31 prevention, prognosis, and management of CAD' should also be toned down. This was a study of a moderate-sized population in a specific geographical area and ethnic subgroup and generalizability of the results described in the article should be very cautious.

- Language should be improved. Multiple minor and major errors detract from the manuscript's value. and probably distract the readers' attention.

Eg, there is no true 'thrombophilia' in CAD (Introduction).

- The definition of CAD is not the one that the authors use in the manuscript and the sentence 'Patients who have experienced myocardial infarction (MI), undergone percutaneous coronary intervention (PCI), or received a coronary artery bypass graft are diagnosed with coronary heart disease (CHD)' describes only a proportion of the patients with CAD while in reality CAD is a much broader entity.

- My major concern is that although there is a statistically significant difference in the polymorphism investigated in these groups, so were classic cardiovascular risk factors between the two cohorts (DM, HTN, obesity, FBG). It may be conceivable that it is difficult to arrange for a matched cohort of persons free of CAD but with no differences in these risk factors, but this essentially means that the results of this study should be interpreted with extreme caution.

Overall, I believe that after heavy editing this paper may merit publication.

Multiple minor and major errors.

Not clear if this is a language-barrier but it becomes evident that the selection of words sometimes is poor.

Author Response

Response to the Reviewer’s comments

Reviewer 2

1) In the Abstract, there is no clue on the thought-process behind selecting this particular gene to investigate the association with CAD. Is should be reported why the authors opted to investigate this particular gene among others.

=> We thank the reviewer for the valuable comments. As you mentioned, we added the sentences that the reasons for selecting the TS gene in association with CAD disease. The added sentences are as follows: “This study aimed to investigate the connection between genetic factors and CAD, focusing on the thymidylate synthase (TS) gene, a gene involved in DNA synthesis and one-carbon metabolism. TS plays a critical role in maintaining the deoxythymidine monophosphate (dTMP) pool, which is essential for DNA replication and repair.”

2) It is unclear why in the first paragraph, in the Discussion section the authors undertook a discussion around association of CAD and stroke; this is quite irrelevant with the study.

=> We thank the reviewer for the critical comments. As you mentioned, we deleted some sentences and modified the first paragraph in the discussion section. The revised paragraph is as follows: “The mortality rate associated with CAD is nearly three times higher than that of stroke, underscoring the importance of screening for CAD and identifying diagnostic markers to improve prognosis [18-20]. Moreover, numerous studies have investigated the prevalence of subclinical cardiovascular disease in ischemic stroke patients using coronary computed tomographic angiography (CTA) and various surrogate markers of systemic atherosclerosis. These studies have explored the relationship between subclinical CAD and vascular risk factors [21]. Therefore, we aimed to identify associations between CAD onset and diagnostic markers. To address this objective, we recognized the necessity to investigate the onset and treatment of the disease. Based on this rationale, we comprehensively analyzed mutations in the TS gene in CAD patients and control subjects.”

3) Endothelial dysfunction is not necessarily a precursor of CAD ('Endothelial dysfunction, a precursor to cardiovascular or cerebrovascular diseases', Discussion section). Endothelial dysfunction could co-exist, predate, antedate or any combination of these.

=> We thank the reviewer for the valuable comments. As you mentioned, we modified the sentence in the discussion section. The revised sentence is as follows: “Endothelial dysfunction initiates a detrimental cycle culminating in overt atherosclerosis, significant CAD, silent brain infarction (SBI), plaque rupture, and ultimately MI or ischemic stroke [42-44].”

4) Some sections need to be toned down.

Eg the causes of coronary artery disease are far from unclear and hence the sentence 'While distinct features of CAD have been identified, its causes remain largely unclear' should be changed or deleted altogether.

Additionally, bold statements like 'we propose that polymorphisms in 30 the 3'-UTR miRNA binding site of the TS gene could serve as clinically useful biomarkers for the 31 prevention, prognosis, and management of CAD' should also be toned down. This was a study of a moderate-sized population in a specific geographical area and ethnic subgroup and generalizability of the results described in the article should be very cautious.

=> We thank the reviewer for the critical comments. As you mentioned, we modified the sentences in the abstract section. The modified sentences are as follows: “While distinct features of CAD have been reported, the association between genetic factors and CAD in terms of biomarkers was insufficient.”, “Based on these findings, we propose that polymorphisms in the TS gene had the possibility of clinically useful biomarkers for the prevention, prognosis, and management of CAD in the Korean population.”

5) Language should be improved. Multiple minor and major errors detract from the manuscript's value. and probably distract the readers' attention.

Eg, there is no true 'thrombophilia' in CAD (Introduction).

=> We thank the reviewer for the valuable comments. As you mentioned, we generally modified the manuscript.

6) The definition of CAD is not the one that the authors use in the manuscript and the sentence 'Patients who have experienced myocardial infarction (MI), undergone percutaneous coronary intervention (PCI), or received a coronary artery bypass graft are diagnosed with coronary heart disease (CHD)' describes only a proportion of the patients with CAD while in reality CAD is a much broader entity.

=> We thank the reviewer for the valuable comments. As you mentioned, we changed the reference in the sentence. The changed reference is as follows: "3.      Di Angelantonio, E.; Thompson, A.; Wensley, F.; Danesh, J., Coronary heart disease. IARC Sci Publ 2011, (163), 363-86.”

7) My major concern is that although there is a statistically significant difference in the polymorphism investigated in these groups, so were classic cardiovascular risk factors between the two cohorts (DM, HTN, obesity, FBG). It may be conceivable that it is difficult to arrange for a matched cohort of persons free of CAD but with no differences in these risk factors, but this essentially means that the results of this study should be interpreted with extreme caution.

=> We thank the reviewer for the valuable comments. As you mentioned, we accepted your concern and considered the possibility of TS gene polymorphism in the CAD biomarkers when interpreting these results.

Round 2

Reviewer 1 Report

According to my comments, the authors seem to have revised their manuscript as much as possible. I have no further comments.

Reviewer 2 Report

I thank the authors for expeditiously reviewing the manuscript and editing all necessary points and addressing all concerns raised.

I now believe this is a more appropriate version for publication.

Looks reasonably ok.